# Exploration of Caregiver Experiences of Conservatively Managed End-Stage Kidney Disease to Inform Development of a Psychosocial Intervention: The ACORN Study Protocol

**DOI:** 10.3390/healthcare9121731

**Published:** 2021-12-14

**Authors:** Claire Carswell, Magdi Yaqoob, Patricia Gilbert, Ying Kuan, Gladys Laurente, Karen McGuigan, Clare McKeaveney, Clare McVeigh, Joanne Reid, Soham Rej, Ian Walsh, Helen Noble

**Affiliations:** 1School of Nursing and Midwifery, Queen’s University Belfast, Belfast BT7 1NN, UK; G.Laurente@qub.ac.uk (G.L.); K.McGuigan@qub.ac.uk (K.M.); C.McKeaveney@qub.ac.uk (C.M.); Clare.McVeigh@qub.ac.uk (C.M.); j.reid@qub.ac.uk (J.R.); helen.noble@qub.ac.uk (H.N.); 2Department of Health Sciences, University of York, York YO10 5DD, UK; 3Barts Health NHS Trust, London E1 1BB, UK; m.m.yaqoob@qmul.ac.uk; 4Northern Ireland Kidney Patient Association, Belfast BT9 7AB, UK; pattyg@btinternet.com; 5Western Health and Social Care Trust, Londonderry BT47 6SB, UK; Ying.Kuan@westerntrust.hscni.net; 6Department of Psychiatry, Jewish General Hospital/Lady Davis Institute, McGill University, Montréal, QC H3A 0G4, Canada; soham.rej@mcgill.ca; 7School of Medicine, Dentistry and Biomedical Sciences, Queen’s University Belfast, Belfast BT7 1NN, UK; i.walsh@qub.ac.uk

**Keywords:** end-stage kidney disease, carers, informal caregiver, psychosocial support, qualitative, conservative management, palliative care

## Abstract

Background: End-stage kidney disease (ESKD) is an overwhelming illness that impacts not just patients, but also their informal carers. Patients who opt for conservative management rather than dialysis experience difficult symptoms and the psychosocial consequences of their condition. Informal carers of patients who choose conservative management can also experience high levels of psychosocial burden, yet there is little guidance on how best to support informal carers, and no evidence on psychosocial interventions to address unmet needs. Aim: The aim of this study is to explore the experiences and unmet needs of informal carers of patients with ESKD receiving conservative management in order to inform the development of a psychosocial intervention. Methods: This qualitative study will consist of three stages: (I) semi-structured interviews with informal carers in England and Northern Ireland, (II) focus groups with healthcare professionals and informal carers, and (III) national workshops to refine the components of a psychosocial intervention. Discussion: Informal carers of patients with ESKD who are receiving conservative management experience a high psychosocial burden, but there is limited evidence on how best to provide support, particularly as the patient nears the end of life. To our knowledge this study will be the first to address this gap by exploring the experiences and unmet needs of informal carers, with the aim of informing the development of a psychosocial intervention to support the health and wellbeing of informal carers.

## 1. Introduction

End-stage kidney disease (ESKD) is an all-consuming condition that significantly impacts the physical and psychological wellbeing of both patients and their informal carers [1,2,3]. Treatment options include renal replacement therapies such as transplant and dialysis; however, these may provide little survival benefit for people who are frail, elderly and with additional co-morbidities [1]. An alternative option to renal replacement therapies is conservative management (CM), where patients choose not to undergo dialysis and instead choose symptom management. Psychosocial and spiritual support are prioritised, treated, and supported by the healthcare team until death [1].

Patients who opt for CM also frequently present with multiple co-morbid conditions, experience a high symptom burden, and have a limited life expectancy and significant unmet palliative care needs [4]. In addition, they may experience a feeling of abandonment by their healthcare team when they choose not to dialyse, underpinned by prognostic uncertainty [5]. Effective management of ESKD is multifaceted, involves complex dietary and medication regimes, and impacts those who are emotionally and/ or practically involved in providing care [6]. Accordingly, informal carers of patients with ESKD receiving CM may have their own health and social care needs that must be addressed [7].

The experiences and perspectives of informal carers of patients with ESKD who are receiving CM, particularly as the patient deteriorates towards death, are generally unknown [8]. In addition, there is a lack of available evidence on supportive interventions for informal carers of patients with ESKD. One reason for this is the fact there is little known about the needs and preferences surrounding psychosocial support for informal carers of patients with ESKD receiving CM, but adjustment to the role is challenging [9]. Most qualitative work among informal carers has been limited to specific treatments, or specific situations such as dialysis discontinuation [10,11,12]. In addition to this lack of an evidence-base, there is little acknowledgement of the role and the needs of informal carers in policy guidance produced by the National Institute for Clinical Excellence [13], National Service Framework [14] or Renal Association Guidelines [15].

As a result of this dearth of evidence and lack of support available for informal carers, the recent international KDIGO Controversies Conference on Supportive Care in Chronic Kidney Disease highlighted the need to address the psychosocial impact of kidney disease on not only patients with ESKD, but also their informal carers, and identify the priorities and needs of informal carers towards the end of life [16].

### Aim

To explore the experiences and unmet needs of informal carers of patients with ESKD receiving CM, and to inform the development of a psychosocial intervention to support them in their caring role.

## 2. Materials and Methods

### 2.1. Study Design

This study will consist of three stages and will follow the Medical Research Council’s (MRC) framework, specifically phase 1 of developing and evaluating complex interventions [17]. The methodology has been informed by O’Cathain et al.’s guidance on how to develop complex interventions [18]. The study will take an interpretivist approach to study design, data collection and analysis. Interpretivism is a philosophical theory that frequently underpins qualitative research, and posits that a person’s social reality is constructed through their own subjective interpretation of their experiences [19]. Consequently, the study’s data collection will be based on understanding the lived experience of informal caregivers of people with ESKD receiving CM, as opposed to explaining, generalizing or critiquing [20].

The study will consist of three stages, as the results of each stage of the study will inform the data collection in the subsequent stage. The first stage of the study will involve semi-structured interviews with informal carers; the second stage will consist of focus groups with relevant healthcare professionals and informal carers; and the third stage will comprise national workshops.

### 2.2. Research Setting

This study will take place across two UK sites, one in Northern Ireland and one in England. Participants from Northern Ireland will be recruited from a renal service within the Health and Social Care Trust that serves a largely rural, white community with an older patient group. Participants from the site in England will be recruited from a specialist CM service provided in a large, urban hospital serving a younger and ethnically diverse population.

### 2.3. Stage 1

Semi-structured interviews with informal caregivers.

#### 2.3.1. Participants

Eligibility criteria for informal carers:-Aged 18 years or over.-Able to give informed consent.-Nominated by the patient as a carer.-Caring for a patient with ESKD, who is receiving CM, and is living at home.

Participants will consist of informal carers of patients with ESKD who are receiving CM, therefore informal carers will be excluded if they are caring for a patient receiving renal replacement therapy. A purposive sampling technique will be used to recruit approximately 60 carers, 30 from the site in Northern Ireland and 30 from the site in England. While participants will be recruited until data saturation is reached, between 30–60 interviews should be an adequate amount of semi-structured interviews to ensure that all relevant data reaches saturation [21].

#### 2.3.2. Recruitment

Initial contact will be made by senior renal nurses who will act as gatekeepers. The research team will brief these nurses on the project prior to recruitment to ensure they are aware of the eligibility criteria and are able to answer any questions patients may have for them. These nurses will identify appropriate patients who are receiving CM for ESKD, explain the purpose of the study, and confirm whether the patient has an informal carer who may be willing to participate. If the patient is eligible and interested they will be provided with a sealed envelope containing a cover letter and an information sheet to be given to their key informal carers. A contact number and email address will be provided for informal carers to call or email if they wish to participate in the study. A self-addressed envelope will also be provided which informal carers may return if they wish to participate in the study if they do not want to call or email.

When informal carers make contact, an initial discussion will be held to answer any queries, and arrange the date, time and location of the interview should they wish to participate. Posters outlining the study will be displayed within renal clinics with information about how to get involved in the study. The study will also be advertised on social media and through patient association websites such as Kidney Care UK and Northern Ireland Kidney Patient Association.

#### 2.3.3. Data Collection

Semi-structured interviews will be conducted to explore the experiences of informal carers of patients with ESKD receiving CM, focusing on their unmet needs and the relationship between their caring role and their psychosocial wellbeing. These interviews will be conducted virtually, over the phone, or in person, depending on the participant’s preferences on location and time. The audio of the interviews will be recorded and transcribed verbatim, with the exception of identifiable data, which will be removed from the final transcripts. The audio recordings, transcripts and personal information of participants will be stored securely on encrypted servers and a data protection impact assessment will be completed to ensure that any risks of data breaches are mitigated.

While it is anticipated that approximately 60 participants will be needed to reach data saturation, data collection will continue until data saturation is reached across the two recruitment sites. The semi-structured interviews will last approximately 60 minutes and consist of questions related to experiences providing care, centered around general experiences of providing care, experiences of ESKD and decision-making related to CM, disease prognosis, and current support and coping strategies. Participants will be provided with information and contact details of support services available, both at the end of the interview and on the participant information sheet provided prior to consent.

### 2.4. Stage 2

Focus groups with healthcare professionals and informal carers.

#### 2.4.1. Participants 

Eligibility criteria for renal healthcare professionals:-Presently working for at least three months within the renal specialty.-Have an appropriate professional qualification and registration (registered nurse, registered doctor, registered dietitian, registered allied healthcare professionals) in their field of work.

Eligibility criteria for informal carers who will participate in the focus groups will be the same as the eligibility criteria as stage 1.

Healthcare professionals will be excluded if they do not have experience in care delivery to patients with ESKD receiving CM, or if they are employed by a recruitment agency or do not hold a permanent healthcare contract. Purposive sampling will be used to recruit a wide range of healthcare staff working in the renal specialty with an interest in CM. Informal carers who take part in the semi-structured interviews will be asked if they are willing to participate in focus groups during the second stage of the study.

Approximately 20 renal HCPs and 12 informal carers will be recruited into stage 2 of the study, across the two UK sites. Two focus groups will take place at each of the two sites.

#### 2.4.2. Recruitment

Recruitment of healthcare professionals will take place after stage one of the study, including data collection, transcription, and analysis is complete. Initial contact with renal healthcare professionals will take place over a number of weeks during weekly clinical governance meetings which are routinely attended by a range of renal healthcare professionals with expertise in supporting patients with kidney disease and their caregivers. A member of the research team will present a brief overview of the study, the progress to date and invite HCPs to take part. Participant information sheets will be provided with contact details of the research team for the HCPs to self-refer if they want to participate. Posters will also be displayed in renal clinics, and social media accounts will be used to advertise the study.

Informal carers who express an interest in participating in focus groups during stage 1 will be asked for consent to contact them for stage 2. 

#### 2.4.3. Data Collection

Each focus group will consist of approximately eight people, with an optimum range of six to eight in order to provide a variety of perspectives without being small enough to be disorganised and fragmented [22]. The participants in each focus group will consist of three informal carers and five HCPs. Findings from Stage 1 of the study will be shared with focus groups participants, allowing the informal carers in the group to answer any questions raised and offer insight into the collected data. As a result, the focus groups will cover similar topics in relation to the general experience of caregiving, the experiences of ESKD, decision making around CM, the role of the professional in providing support to informal caregivers, prognosis of the disease, current support and coping strategies. The discussions that come from these focus groups will build on the findings from Stage 1 and provide a more comprehensive overview of the experiences of informal caregivers in the context of the healthcare service.

Telephone and video conferencing will be offered to enable participants to take part, particularly informal carers who may find it difficult to travel long distances. Focus groups will be conducted remotely or through video conferencing if national or regional restrictions make in person focus groups prohibitive. The focus groups will last for approximately 90 min, and participants will be provided with information and contact details of support services available, both at the end of the interview and on the participant information sheet provided prior to consent. The audio of the focus groups will be recorded and transcribed verbatim, with the exception of identifiable data, which will be removed from the final transcripts. The audio recordings, transcripts and personal information of participants will be stored securely on encrypted servers and a data protection impact assessment will be completed to ensure that any risks of data breaches are mitigated.

### 2.5. Stage 3

National workshops will be held across the two sites, with one workshop conducted in Northern Ireland and the other in England, with key stakeholders, and will be facilitated by experts in the field, including representatives from Kidney Care UK and service users. These workshops will be offered online and face-to-face to maximise participation by increasing accessibility. The findings from Stage 1 and 2 will be presented at the workshops in order to identify modifiable factors contributing to problems of informal caregivers and explore potential mechanisms through which change could occur and how these mechanisms could be delivered.

Breakout sessions will be facilitated, with delegates split into smaller groups and asked to share their views on support for informal caregivers, the proposed problems/challenges, mechanisms and systems of delivery through which support could be provided to informal caregivers. These breakout sessions will be recorded and transcribed verbatim.

Core components of an intervention will be identified during the breakout sessions at the national workshops. These findings will inform future intervention development work that will formalise a psychosocial intervention to support carers of patients with ESKD receiving CM.

### 2.6. Data Analysis

The transcribed qualitative data from Stages 1, 2, and 3 will be analysed using the thematic content analysis guided by King and Horrocks [23]. This aligns with the interpretivist approach of the study and will allow for an inductive approach to the identification of the components of a potential psychosocial intervention. The first step will involve descriptive coding of the data line by line, identifying words or phrases that capture salient components within the data. During the second step, interpretive codes will be synthesized from the descriptive codes. The interpretive codes will be arranged into hierarchical categories, forming final overarching themes [24]. Thematic analysis involves the detection of themes recurring in that data, and will incorporate open coding to organise and assimilate the raw data in order to gain an understanding of the multifaceted views, opinions and experiences of the participants. Data will be coded by two researchers and will be managed using NVivo version 11 [25]. An audit trail will be kept in NVivo of the coding process and the development of themes and subthemes.

The credibility of the findings of the study will be evaluated through a number of main methods [26]. Firstly, source triangulation will occur throughout the study by including healthcare professionals and informal caregivers during data collection from two different settings. Secondly, evaluation will be undertaken by respondent validation, both during the workshops, where the findings from the semi-structured interviews and focus groups will be presented to key stakeholders, including respondents from the earlier stages of the study, and through ongoing member-checking during the analysis project. Thirdly, research team members will provide feedback consistently on emerging themes and subthemes to challenge assumptions and reach consensus on the analysis. 

## 3. Discussion

The support needs of informal carers of patients with ESKD receiving CM are largely unknown. Previous qualitative research that has explored the experiences of this population highlighted that they may have gaps in knowledge around what CM means [10] and the difficulties surrounding the decision making process when choosing between dialysis and CM [27]. However, most research has focused on the experiences of patients receiving CM, and as a result there is little understanding of how caring for a person with ESKD receiving CM may impact on an informal carer’s psychosocial wellbeing and needs. Therefore, this study will provide a growing evidence base which has, for this group, been lacking.

The main strength of this study is the use of a robust interpretivist qualitative approach that focuses on the needs and experiences of informal carers for patients with ESKD receiving CM. This will provide a rigorous foundation on which to develop a supportive intervention for this population. However, there are a number of limitations, including the lack of generalisability, that arise from this study. While we aim to achieve a more diverse sample by conducting this qualitative study in two separate locations within the UK, the transferability of our findings may be limited on a more international scale due to differences between countries in relation to healthcare systems, access to CM and treatment for CKD, and different cultures that influence traditional caring roles.

## 4. Conclusions

The proposed study will elucidate the experiences of informal carers of patients with ESKD receiving CM and help identify their unmet support needs in order to inform the future development of a supportive intervention. Little is known about the experiences of informal carers of patients with ESKD receiving CM, therefore this research will provide novel insights that can inform future research and adjustments to practice beyond the scope of this study alone.

## Data Availability

Not applicable.

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
