# Peer review of "Exploration of Caregiver Experiences of Conservatively Managed End-Stage Kidney Disease to Inform Development of a Psychosocial Intervention: The ACORN Study Protocol"

_healthcare, 2021, doi:10.3390/healthcare9121731_

Round 1

Reviewer 1 Report

The manuscript is a protocol for a study. However, it is nowhere mentioned in the title and abstract. It is important that the study protocol should be clear from the title itself. The only inference that we get from the title and abstract is 'the study will consist of three stages'.

The introduction is well written. The authors have provided an excellent summary of the need for this study.

There are three stages in the study. The first phase has a systematic review and semi-structured interviews (SSI) with caregivers. The second phase has FGD with healthcare workers and caregivers. It is unclear from the manuscript what is the aim of having this second stage? Why not include SSI and FGD as a single entity? Why the authors are conducting SSI and FGDs with the same caretakers but at different stages of the study? It needs a detailed explanation of the rationale behind the different stages of the study.

There is no link between systematic review and SSI. What is the role of the systematic review in this study?

The word healthcare professionals are broad. It can include anyone involved in healthcare practice. It needs to be clear who are the healthcare professionals, for example, doctors, nurses etc. Also, the use of abbreviations for academic degrees should be avoided (eg. RGN)

The authors mention that FGDs may be conducted remotely depending on national or regional restrictions. However, they are not clear for SSI.

The authors need to describe how they will translate the findings from the study into the development of a psychosocial intervention.

Author Response

Thank you so much for your feedback and taking the time to review the protocol.

The manuscript is a protocol for a study. However, it is nowhere mentioned in the title and abstract. It is important that the study protocol should be clear from the title itself. The only inference that we get from the title and abstract is 'the study will consist of three stages'.

The title now reads 'The ACORN study protocol' 

The introduction is well written. The authors have provided an excellent summary of the need for this study.

Thank you very much!

There are three stages in the study. The first phase has a systematic review and semi-structured interviews (SSI) with caregivers. The second phase has FGD with healthcare workers and caregivers. It is unclear from the manuscript what is the aim of having this second stage? Why not include SSI and FGD as a single entity? Why the authors are conducting SSI and FGDs with the same caretakers but at different stages of the study? It needs a detailed explanation of the rationale behind the different stages of the study.

The reason why these two stages are being considered as two distinct entities is because the results of the first stage of the study (the semi-structured interviews in this case) are being used to inform the data collected from the focus groups. The reference to this has now been highlighted in the data collection section for the second stage, and this has been clarified in the overall design of the study.  

There is no link between systematic review and SSI. What is the role of the systematic review in this study?

As the protocol for the systematic review will be published separately and is in the process of being registered in PROSPERO we have removed it from the protocol. The systematic review will inform the national workshops in the third stage of the study, however this appears to be muddying the waters of the protocol for the qualitative aspects of the study so we have removed reference to it from the manuscript.  

The word healthcare professionals are broad. It can include anyone involved in healthcare practice. It needs to be clear who are the healthcare professionals, for example, doctors, nurses etc. Also, the use of abbreviations for academic degrees should be avoided (eg. RGN)

We are including anyone who is considered a healthcare professional according to qualification and professional registration, we have provided an overview of some examples without using abbreviations. 

The authors mention that FGDs may be conducted remotely depending on national or regional restrictions. However, they are not clear for SSI.

This is clarified in the highlighted section under data collection for Stage 1.

The authors need to describe how they will translate the findings from the study into the development of a psychosocial intervention.

This is outside of the scope of the study. This study is to inform the development of a psychosocial intervention by identifying core components, but not to develop a full psychosocial intervention - I have highlighted the section of the manuscript that outlines this, and the development of a psychosocial intervention will hopefully be the focus of future related research. 

Reviewer 2 Report

This protocol lists steps for a study consisting of a systematic review (SR) of the literature and a qualitative analysis of semi-structured interviews with informal caregivers of conservatively managed end-stage kidney disease patients and focus groups of healthcare professionals and informal caregivers, at the end adding national workshops. The following points will help strengthen the manuscript.

1. Title: Could authors revise the title indicating that this is a protocol (more specifically, it should also indicate that this is a SR protocol; please see below)?
2. Title: Could authors explain why “Caregivers” and “Study” in the title start with capital letters while the rest is in lower-case?
3. P.2 Materials and Methods: Thanks for the information. Please consider separating the steps in this protocol into the conduct of the SR and the qualitative study (as an initial review point, it is important to be very clear why focus groups follow semi-structured interviews in the case of informal caregivers in this qualitative study and it is not the other way around and also it would be helpful to be very clear why both data collection strategies are needed) and prepare two separate protocols. At this phase, this reviewer will just consider that this is a protocol for the SR of the literature part of the study and suggest review points accordingly.
4. P. 1-2: Please revise the Abstract and Introduction sections motivating why a SR is needed on this topic. What is known about the subject matter so far? Are there other literature reviews, SRs, scoping reviews etc. on the topic? What are their findings? Please be clear what are the gaps in the literature and why a new SR is needed.
5. P.2 Introduction: Please ensure that the aim of this protocol is clearly stated in terms of the SR that will be conducted.
6. P.2-3 Materials and Methods: Thanks for indicating “PROSPERO”. What is the PROSPERO registration number?
7. P.2-3 Materials and Methods: Thanks for indicating “PRISMA”. Which guideline was followed in developing this protocol?
8. P.2-3 Materials and Methods: Please clearly define the population, intervention, control and outcomes that will be considered for this SR providing concise definitions of all the concepts used and incorporating sufficient detail about the inclusion and exclusion criteria (so that a replication of the SR will be possible if warranted).
9. P.2-3 Materials and Methods: Please indicate what types of study designs will be included. Define the study designs that will be included fully.
10. P.2-3 Materials and Methods: Which databases, websites etc.. will be searched? Please provide complete names of the information sources.
11. P.2-3 Materials and Methods: Will the assistance of a librarian be available in database searches?
12. P.2-3 Materials and Methods: Please provide a preliminary copy of a search strategy for one database in an appendix. When was this preliminary search conducted? Did this initial search produce reasonable results? 
13. P.2-3 Materials and Methods: Please provide the expertise of the team as it relates to the subject matter of this SR.
14. P.2-3 Materials and Methods: Please provide other information about the SR such as the time horizon that will be considered in the searches, languages that will be included etc…
15. P.2-3 Materials and Methods: Please indicate the specifics of the selection processes that will used in different phases of the SR such as the screening phase, full-text review phase…
16. P.2-3 Materials and Methods: Please provide information about data extraction forms. If the forms are being developed for this SR, how will they be tested….
17. P.2-3 Materials and Methods: Please provide information about data items that will be collected.
18. P.2-3 Materials and Methods: Please indicate how risk of bias will be assessed across studies.
19. P.2-3 Materials and Methods: Please indicate how the results of this SR will be synthesized.

Author Response

Thank you so much for your feedback

1. Title: Could authors revise the title indicating that this is a protocol (more specifically, it should also indicate that this is a SR protocol; please see below)?

Protocol has been added to the title, it now reads 'The ACORN study protocol'

2. Title: Could authors explain why “Caregivers” and “Study” in the title start with capital letters while the rest is in lower-case?

This was left over from the capitalisation to inform acronyms. This has been corrected and it is now in sentence case.

3. P.2 Materials and Methods: Thanks for the information. Please consider separating the steps in this protocol into the conduct of the SR and the qualitative study (as an initial review point, it is important to be very clear why focus groups follow semi-structured interviews in the case of informal caregivers in this qualitative study and it is not the other way around and also it would be helpful to be very clear why both data collection strategies are needed) and prepare two separate protocols. 

Thank you for this feedback, we have removed reference to the systematic literature review from the protocol. As it is still under review for registration with PROSPERO this protocol now only relates to the qualitative portions of the study. 

Round 2

Reviewer 1 Report

The protocol now is more clear than the previous version. 

All the concerns have been well addressed. The removal of the section of the systematic review adds clarity to the protocol.

Author Response

All the concerns have been well addressed. The removal of the section of the systematic review adds clarity to the protocol.

Thank you very much for your feedback and comments!

Reviewer 2 Report

Previous protocol was listing steps for a study consisting of a systematic review (SR) of the literature and a qualitative analysis of semi-structured interviews with informal caregivers of conservatively managed end-stage kidney disease patients and focus groups of healthcare professionals and informal caregivers, at the end adding national workshops. This reviewer had initially suggested that authors should start with a protocol for SR (given that a literature review of any type is a natural starting point for any study) and added numerous review points for SR. Rather than responding to that initial review, authors, in this version, have excluded the SR part of the protocol, therefore did not respond to any of the review points and just left in the qualitative analysis section, making it a new review from the perspective of this reviewer. The following (new) points will help strengthen the qualitative analysis part of the original manuscript.

1. Title: Thanks for adding ‘protocol’ to the title. Similar to the previous review, why is protocol in a capital letter while all else is in lower-case letters?
2. P.2 Materials and Methods: Please provide a rationale for choosing “interpretivist approach to study design”. How is this study design different from other qualitative study designs? Please explain.
3. P.2 Materials and Methods: Could authors provide further information about the characteristics of these 2 settings?
4. P.3 Materials and Methods, Semi-structured interviews: How did authors decide on these sample sizes? How is it possible to know ahead of time that saturation will be achieved with these sample sizes?
5. P.3 Materials and Methods, Semi-structured interviews: Please provide a copy of an initial semi-structured interview guide.
6. P.3 Materials and Methods, Semi-structured interviews: Will there be a pilot testing of this initial semi-structured interview guide?
7. P.3 Materials and Methods, Semi-structured interviews: Please provide information about these semi-structured interviews such as how many questions will be included, how long are interviews expected to last, will there be debriefing after interviews….
8. P.3 Materials and Methods, Semi-structured interviews: How will the privacy and confidentiality of information collected during these interviews be secured?
9. P.3 Materials and Methods, Semi-structured interviews: Who will conduct these interviews? How many interviewers will there be?
10. P.3-4 Materials and Methods, Focus Groups: Please provide information about these focus groups such as how many questions will be included, how long are these focus groups expected to last, will there be debriefing after focus groups….
11. P.3-4 Materials and Methods, Focus Groups: How will authors approach the privacy and confidentiality issues in the conduct of these focus groups?
12. P.3-4 Materials and Methods, Focus Groups: Please discuss how the 3 informal caregivers who are planned to attend each of the focus groups will represent informal caregivers recruited for this study (or be representative in general) if this qualitative study depends on a design where the “philosophical theory” that “underpins…posits that a person’s social reality is constructed through their own subjective interpretation of their experiences”?
13. P.3-4 Materials and Methods, Focus Groups: What measures are in place to ensure that data from the focus group discussions will be appropriately collected/depicted?
14. P.5 Materials and Methods, Data Analysis: How many researchers and who will conduct the coding etc?
15. P.5 Materials and Methods, Data Analysis: What procedures will be in place to ensure the credibility of findings?
16. P.5 Materials and Methods, Data Analysis: Will there be an audit trail available?
17. P.5 Discussion: Please discuss if findings from a qualitative study can be generalized? Please tie to the above point about the “philosophical theory” that authors posit as underpinning this study design or qualitative studies in general.

Author Response

Thank you so much for taking the time to review our manuscript and providing such valuable constructive feedback, this has been tremendously helpful.

Title: Thanks for adding ‘protocol’ to the title. Similar to the previous review, why is protocol in a capital letter while all else is in lower-case letters?

This has been edited and is now lower case.

 P.2 Materials and Methods: Please provide a rationale for choosing “interpretivist approach to study design”. How is this study design  different from other qualitative study designs? Please explain.

This informs the general aim and approach to data collection, in terms of aiming to understand the experiences as opposed to explaining the experiences, critiquing or generalising the experiences. A sentence has been added to this effect. 

P.2 Materials and Methods: Could authors provide further information about the characteristics of these 2 settings?

Specifics on the characteristics of the two settings have now been provided.

4. P.3 Materials and Methods, Semi-structured interviews: How did authors decide on these sample sizes? How is it possible to know ahead of time that saturation will be achieved with these sample sizes?

An explanation for the sample size is provided, and it is now clarified that data will be collected until data saturation is reached. As the ethical application requires a specific number of participants we had to specify the sample size. 

P.3 Materials and Methods, Semi-structured interviews: Please provide a copy of an initial semi-structured interview guide.

This has now been attached as material for review. 

6. P.3 Materials and Methods, Semi-structured interviews: Will there be a pilot testing of this initial semi-structured interview guide?

The semi-structured interviews have been based on the interview guides for a similar study for carers of people receiving haemodialysis. The interview guides have been reviewed by our patient and public involvement group to ensure they're appropriate.

7. P.3 Materials and Methods, Semi-structured interviews: Please provide information about these semi-structured interviews such as how many questions will be included, how long are interviews expected to last, will there be debriefing after interviews….

These details have now been added to the 'data collection' section for Stage 1.

8. P.3 Materials and Methods, Semi-structured interviews: How will the privacy and confidentiality of information collected during these interviews be secured?

This has now been provided for both the semi-structured interviews and the focus groups, under the section on data collection

9. P.3 Materials and Methods, Semi-structured interviews: Who will conduct these interviews? How many interviewers will there be?
10. P.3-4 Materials and Methods, Focus Groups: Please provide information about these focus groups such as how many questions will be included, how long are these focus groups expected to last, will there be debriefing after focus groups….

This has now been added to the section of data collection within stage 2. 

11. P.3-4 Materials and Methods, Focus Groups: How will authors approach the privacy and confidentiality issues in the conduct of these focus groups?

A statement on confidentiality and privacy has been added to the data collection section that relates to focus groups.

12. P.3-4 Materials and Methods, Focus Groups: Please discuss how the 3 informal caregivers who are planned to attend each of the focus groups will represent informal caregivers recruited for this study (or be representative in general) if this qualitative study depends on a design where the “philosophical theory” that “underpins…posits that a person’s social reality is constructed through their own subjective interpretation of their experiences”?

The informal caregivers are not intended to be representative of the entire population of informal caregivers, nor the population of caregivers interviewed during the first stage, as qualitative research is not intended generalisable. Discussion of representativeness of the sample would not fit into a qualitative manuscript as representative findings that can be generalised are not in line with the aims of the study, nor do they align with the philosophy that has been described, as already stated here. 

13. P.3-4 Materials and Methods, Focus Groups: What measures are in place to ensure that data from the focus group discussions will be appropriately collected/depicted?

This has now been clarified in the manuscript - the focus groups will be audio recorded and transcribed, with identifiable information removed from the final transcripts. 

14. P.5 Materials and Methods, Data Analysis: How many researchers and who will conduct the coding etc?

Data will be coded by two researchers, I have not specified the identity of these two researchers just to mitigate for the risk of staff changes to the research team during the conduct of the study. 

15. P.5 Materials and Methods, Data Analysis: What procedures will be in place to ensure the credibility of findings?

A section on credibility has been added to the data analysis section.

16. P.5 Materials and Methods, Data Analysis: Will there be an audit trail available?

Yes, an audit trail will be kept through NVivo, this has been clarified in the section on analysis. 

17. P.5 Discussion: Please discuss if findings from a qualitative study can be generalized? Please tie to the above point about the “philosophical theory” that authors posit as underpinning this study design or qualitative studies in general.

The issue of generalisability is already mentioned in the discussion in relation to the limitations of the study, as this is a qualitative piece of research it cannot, therefore, be generalised. This statement has been highlighted in the manuscript. The philosophical theory used in this study underpins many qualitative studies, however there are other philosophical theories that qualitative research can use that still do not result in generalisability (for example, critical realism). 
